# Cefiderocol Protects against Cytokine- and Endotoxin-Induced Disruption of Vascular Endothelial Cell Integrity in an In Vitro Experimental Model

**DOI:** 10.3390/antibiotics11050581

**Published:** 2022-04-26

**Authors:** Dagmar Hildebrand, Jana Böhringer, Eva Körner, Ute Chiriac, Sandra Förmer, Aline Sähr, Torsten Hoppe-Tichy, Klaus Heeg, Dennis Nurjadi

**Affiliations:** 1Medical Microbiology and Hygiene, Department of Infectious Diseases, Heidelberg University Hospital, 69120 Heidelberg, Germany; s0jaboeh@uni-bonn.de (J.B.); eva.koerner@med.uni-heidelberg.de (E.K.); sandra.foermer@med.uni-heidelberg.de (S.F.); aline.saehr@med.uni-heidelberg.de (A.S.); klaus.heeg@med.uni-heidelberg.de (K.H.); 2Hospital Pharmacy, Heidelberg University Hospital, 69120 Heidelberg, Germany; ute.chiriac@med.uni-heidelberg.de (U.C.); torsten.hoppe-tichy@med.uni-heidelberg.de (T.H.-T.); 3Department of Infectious Diseases and Microbiology, University of Lübeck, 23538 Lübeck, Germany

**Keywords:** cefiderocol, beta-lactam antibiotic, endothelial barrier, vascular leakage

## Abstract

The severe course of bloodstream infections with Gram-negative bacilli can lead to organ dysfunctions and compromise the integrity of the vascular barrier, which are the hallmarks of sepsis. This study aimed to investigate the potential effect of cefiderocol on the barrier function of vascular endothelial cells (vECs) in an in vitro experimental set-up. Human umbilical vein cells (HUVECs), co-cultured with erythrocyte-depleted whole blood for up to 48 h, were activated with tumor necrosis factor-alpha (TNF-α) or lipopolysaccharide (LPS) to induce endothelial damage in the absence or presence of cefiderocol (concentrations of 10, 40 and 70 mg/L). The endothelial integrity was quantified using transendothelial electrical resistance (TEER) measurement, performed at 0, 3, 24 and 48 h after stimulation. Stimulation with TNF-α and LPS increased the endothelial permeability assessed by TEER at 24 and 48 h of co-culture. Furthermore, cefiderocol reduces interleukin-6 (IL-6), interleukin-1β (IL-1β) and TNF-α release in peripheral blood mononuclear cells (PBMCs) following LPS stimulation in a dose-dependent manner. Collectively, the data suggest that cefiderocol may have an influence on the cellular immune response and might support the maintenance of vEC integrity during inflammation associated with infection with Gram-negative bacteria, which warrants further investigations.

## 1. Introduction

Although the effect of antimicrobial therapy on infecting pathogens is clearly understood, the impact of antimicrobial substances on the biological processes of the host is often understudied. There is increasing evidence that antimicrobial substances exhibit a variety of biological actions in addition to their antibacterial action [1,2].

As a novel siderophore-conjugated cephalosporin antimicrobial substance with stability against hydrolysis due to metallo-beta-lactamase produced by Gram-negative bacilli, cefiderocol is a promising agent for the therapy of infections caused by multidrug-resistant Enterobacterales [3]. 

However, in a phase-three clinical trial comparing the efficacy of cefiderocol versus the best available therapy for infection caused by carbapenem-resistant Gram-negative bacilli (CREDIBLE-CR), the all-cause mortality in the cefiderocol group was reported to be higher than in the best therapy group [4]. A significant cause of mortality in infection-associated sepsis is mediated by pathophysiological processes related to vascular permeability [5]. Since resistance towards cefiderocol was not the likely explanation for this observation, we sought to investigate whether cefiderocol displays immunomodulatory properties using an in vitro experimental model. To this end, we investigated the effect of cefiderocol on experimentally (in vitro) induced vascular damage due to bacterial antigens and tumor necrosis factor-alpha (TNF-α) using human umbilical vein cells (HUVECs) and erythrocyte-depleted whole blood (edWB) in a co-culture set-up.

## 2. Results

### 2.1. Cefiderocol Dampens the Release of Pro-Inflammatory Cytokines in PBMCs

First, we investigated whether cefiderocol influences the release of pro-inflammatory cytokines known to cause endothelial damage leading to vascular leakage. To this end, the release of TNF-α, interleukin-6 (IL-6) and interleukin-1β (IL-1β) were quantified by enzyme-linked immunosorbent assay (ELISA) from supernatants of peripheral blood mononuclear cells (PBMCs) stimulated using lipopolysaccharide (LPS) in the presence and absence of cefiderocol. LPS markedly induced IL-6, IL-1β and TNF-α (*p* = 0.02, *p* = 0.05 and *p* = 0.3, respectively). While 10 mg/L cefiderocol had no significant effect and 40 mg/L cefiderocol only significantly reduced the release of IL-1β (*p* = 0.05) but not IL-6, at a concentration of 70 mg/L, the release of both IL-1β and IL-6 were significantly reduced (*p* = 0.05 and *p* = 0.04, respectively). LPS-induced TNF-α was reproducibly but not significantly downregulated (Figure 1). Importantly, the cell viability was not significantly affected by cefiderocol (Appendix A).

### 2.2. Cefiderocol Protects against Cytokine- and Endotoxin-Induced Loss of vEC Integrity in a Dose-Dependent Manner

Next, to determine the relevance of our findings in the context of endotoxin-induced endothelial dysfunction, we performed further experiments using a HUVEC-edWB co-culture model. HUVECs were grown to a confluent monolayer to mimic the blood vessel that is constantly in contact with blood and blood cells, as previously published [6]. For the co-culture, red blood cells were depleted from fresh whole blood to prevent high viscosity in the co-culture set-up and to prevent heavy hemolysis, which may interfere with the performance of further analysis of the supernatant. The vascular endothelial cell (vEC) integrity was assessed by measuring the transendothelial electrical resistance (TEER) of the HUVEC monolayer [6,7]. The loosening of the tight monolayer structure and loss of integrity results in a decrease in TEER.

The vEC permeability was induced using 100 ng/mL TNF-α or 100 ng/mL LPS, known stimuli to disrupt vEC integrity [8,9], in the presence and absence of cefiderocol (c = 70 mg/L) for 48 h. TEER was measured after 0, 3, 24 and 48 h. TNF-α significantly decreased resistance after 24 and 48 h (*p* = 0.05 and *p* = 0.05, respectively), indicating compromised integrity of the monolayer (Figure 2A). Similar observations were made for LPS (*p* = 0.05 for 24 h and *p* = 0.05 for 48 h). No significant effect of cefiderocol on unstimulated HUVECs was detected. Interestingly, cefiderocol protected against LPS- and TNF-α-mediated loss of integrity significantly after 24 h (*p* = 0.05 for both LPS and TNF-α) and 48 h (*p* = 0.05 for both LPS and TNF-α) (Figure 2A).

The protective effect of cefiderocol against endotoxin- and cytokine-induced vascular dysfunction was dose-dependent. While 40 mg/L could still significantly dampen the TNF-α- and LPS-mediated decrease in HUVECs integrity after 48 h (*p* = 0.05 for both LPS and TNF-α), 10 mg/L only had a minor, statistically non-significant effect, indicating a concentration dependency of the protection (Figure 2B).

## 3. Discussion

Increased vascular permeability contributes to the pathophysiology of bacteremia-related complications and is associated with severe organ dysfunction and septic shock [10]. Bacterial endotoxins, such as LPS, can compromise the endothelial integrity, leading to vascular leakage through the induction of inflammatory mediators, such as TNF-α and IL-1β [8,9,11,12].

Our data show that cefiderocol (in the concentrations 40 mg/L and 70 mg/L) is able to reduce the release of proinflammatory cytokines from LPS-activated PBMCs and dampen LPS-mediated vEC permeability. This suggests that the protective effect may be mediated by the suppression of cytokine release. Furthermore, cefiderocol dampened the TNFα-induced loss of vEC integrity, which points towards a further mechanism. Other β-lactams were shown to covalently bind and impair IFNγ [13,14]. Whether this also accounts for cefiderocol and TNF-α or whether the drug effects molecules in vECS cannot be answered with our study. To the best of our knowledge, data on the immunomodulatory effects of beta-lactam antibiotics are limited. Nonetheless, a similar secondary effect for cefazolin (first-generation cephalosporin) has been recently reported [15]. However, we do believe that this secondary effect is not a universal property of beta-lactams/cephalosporins. In a study by Cui et al., it was demonstrated that ceftazidime (third-generation cephalosporin) enhances TNF-α production due to cellular decay, whereas imipenem (carbapenem) did not alter the TNF-α production in macrophages [16]. In line with our findings, antibiotics of the macrolide group (e.g., erythromycin and roxithromycin) have also been demonstrated to inhibit LPS-induced vascular leakage in rat trachea [17,18].

It is acknowledged that the plasma concentration of beta-lactam antibiotics fluctuates over time [19], with the highest concentrations immediately after dose administration [19,20]. This also applies to cefiderocol. In a study on cefiderocol plasma concentrations after a single-dose administration, the c_max_ (maximum plasma concentration) ranges from 7.8 mg/L to 156 mg/L after single IV infusions of 100 mg and 2000 mg, respectively [21]. Based on these data, a half-life of around 2 h for cefiderocol [20] and the recommended dose regimen for cefiderocol of 2000 mg every 8 h by infusion, we have chosen 70 mg/L (half of the c_max_ after single-dose (2000 mg) administration over 60 min by infusion, ~78 mg/L) as the highest concentration for our in vitro experimental set-up with the assumption that at least 70 mg/L of cefiderocol plasma concentration is maintained for ~50% of the time between administrations. The lowest concentration was chosen based on four times the MIC as a target minimum concentration close to 16 mg/L (based on the EUCAST breakpoint v11.0 for cefiderocol of ≥2 mg/L). In the study of Saisho et al., the pharmacokinetics of cefiderocol was determined after a single-dose administration over a 60-min infusion. For clinical use, cefiderocol is approved as a 3-h infusion, so one would expect that the c_max_ would be higher over a longer period due to the prolonged infusion. Our in vitro experimental data implied that the inhibitory effect of cefiderocol on the pro-inflammatory cytokine release was dose-dependent. Thus, a similar, if not stronger, inhibitory effect would be expected after prolonged administration. However, further confirmatory in vivo experiments would be needed to validate this hypothesis. At a concentration of 40 mg/L, we already observed a protective effect of cefiderocol on endotoxin-induced increased endothelial permeability, indicating that high plasma concentration may not be essential for this effect. Our findings may have clinical relevance for the optimization of antibiotic therapy regimens, e.g., via prolonged or continuous infusions, based on therapy drug monitoring (TDM) [22]. Usually, high plasma concentrations and high dosing regimens of antimicrobial substances go hand in hand with more adverse reactions so that TDM-based antibiotic therapy regimens focus on achieving an adequate plasma concentration while maximizing the time over the minimum inhibitory concentration (MIC) at the same time [23].

As previously mentioned, the all-cause mortality in the cefiderocol treatment group was higher than the control group in the CREDIBLE-CR clinical trial [4]. Although our data imply that cefiderocol exhibits immunomodulatory properties in vitro, which may play a role in the outcome of infection following cefiderocol treatment, in vivo validation is still needed. Nonetheless, our data add to the body of evidence that antimicrobial substances may exhibit secondary effects on the host immunity, which warrants further investigations. 

## 4. Materials and Methods

### 4.1. Erythrocytes-Depleted-Whole-Blood (edWB-Blood) Retrieval

Red blood cell lysis (RBC)-buffer (BD Bioscience, Heidelberg, Germany) was added to heparinized fresh blood from healthy donors and incubated for 15 min at room temperature. After centrifugation (1300 rpm, 10 min), supernatant was discarded and the pellet was resuspended in endothelial cell growth medium (PromoCell, Heidelberg, Germany).

### 4.2. Cell Culture

HUVECs (PromoCell, Heidelberg, Germany) were seeded (3 × 10^4^ cells per well) in endothelial cell growth medium (PromoCell, Heidelberg, Germany) on the filter of collagen-coated transwells (Corning, Kaiserslautern, Germany) for eight days (37 °C and 5% CO_2_) until a confluent monolayer was formed. Subsequently, edWB was added to the endothelial cell culture. PBMCs were isolated from blood of healthy donors by density gradient centrifugation (Biocoll separating solution, 1.077 g/mL, Anprotec, Bruckberg, Germany). Cells (1 × 10^6^/^mL^) were seeded in RPMI1640 (Anprotec, Bruckberg, Germany)/10% FCS (Anprotec, Bruckberg, Germany) at 37 °C and 5% CO_2_.

### 4.3. Enzyme-Linked Immunosorbent Assay (ELISA)

Supernatant of PBMCs was used for quantification of released IL-1β, IL-6 and TNFα by ELISA following the manufacturer’s instructions (BD OptEIA ELISA Sets; BD Biosciences Pharmingen, Heidelberg, Germany). Absorbance was measured on a SUNRISE Absorbance reader (Tecan, Salzburg, Austria) and analyzed with Magellan software.

### 4.4. Transendothelial Electric Resistance (TEER) Measurement

HUVECs/edWB co-culture was stimulated with 100 ng/mL LPS (Invivogen, Toulouse, France) or 100 ng/mL TNFα (Bio-Techne, Minneapolis, MN, USA) with and without cefiderocol (Shionogi B.V., Amsterdam, The Netherlands) in a concentration of 10, 40 and 70 mg/L. TEER was measured with a volt-ohm meter EVOM3 and an STX1 electrode (WPI, Friedberg, Germany), as previously described [6] and following the manufacturer’s protocol, after 0 h, 3 h, 24 h and 48 h. Data were calculated by subtracting resistance values of a trasunswell without cells and multiplied with the surface area (1.12 cm^2^) of the transwell.

### 4.5. Statistical Analysis

Statistical analysis was performed by Graphpad Prism v9 (USA). Values were presented as mean with standard deviation. Comparisons of quantitative variables between groups were calculated using the non-parametric Mann–Whitney U test. The level of significance was set at *p*-value < 0.05. In the main text, *p* = 0.0495 was rounded to *p* = 0.05 and was considered as statistically significant.

## 5. Conclusions

Our in vitro investigation implied that cefiderocol may have immunodomulatory properties, which warrants further in vivo validation and investigation to elucidate the underlying mechanisms. The effect of antimicrobial substances on biological functions and their impact on immune response should be better investigated to improve and optimize strategies for rational antibiotic use to help combat antimicrobial resistance.

## Figures and Tables

**Figure 1 antibiotics-11-00581-f001:**
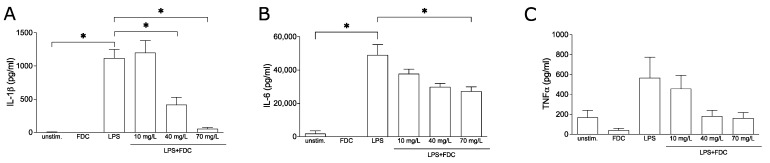
**Cefiderocol dampens LPS-stimulated production of pro-inflammatory cytokines.** PBMCs were stimulated with 100 ng/mL LPS with or without cefiderocol (FDC 10 mg/L, 40 mg/L, 70 mg/L) for 3 days. Unstimulated PBMCs were incubated with 70 mg/L FDC as negative controls. Supernatant was used for ELISA analysis of IL-1β (**A**), IL-6 (**B**) and TNF-α (**C**). Shown are mean and standard deviation of three biological replicates (*n* = 3). Statistical significance was calculated using a one-sided Mann–Whitney U test (*p* < 0.05 = *). Abbreviations: FDC = cefiderocol, LPS = lipopolysaccharide, TNF-α = tumor necrosis factor-α.

**Figure 2 antibiotics-11-00581-f002:**
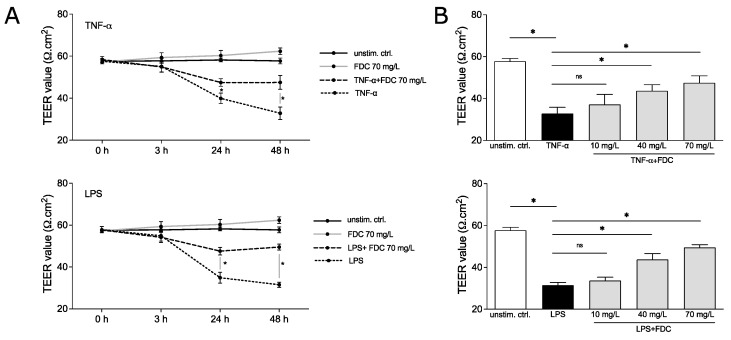
**Cefiderocol dampens LPS and TNF-α induced decrease in endothelial cell integrity.** HUVEC/edWB coculture was stimulated with 100 ng/mL LPS or 100 ng/mL TNF-α with or without cefiderocol (concentrations: (**A**): 70 mg/L, (**B**): 10 mg/L, 40 mg/L, 70 mg/L). TEER was measured with the EVOM3 volt/ohmmeter (timepoints: (**A**): after 0 h, 3 h, 24 h and 48 h, (**B**): after 48 h). Shown are mean and standard deviation of three biological replicates (*n* = 3). Statistical significance was calculated using a one-sided Mann–Whitney U test (*p* < 0.05 = * and was considered statistically significant, ns = statistically not significant). Abbreviations: TEER = transendothelial electrical resistance, FDC = cefiderocol, TNF-α = tumor necrosis factor-α, LPS = lipopolysaccharide.

## Data Availability

Not applicable.

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
