# Peer review of "Cefiderocol Protects against Cytokine- and Endotoxin-Induced Disruption of Vascular Endothelial Cell Integrity in an In Vitro Experimental Model"

_antibiotics, 2022, doi:10.3390/antibiotics11050581_

Round 1

Reviewer 1 Report

  1. The authors should consider adding 1-2 sentences explaining the rationale for investigating cefiderocol specifically. 
  2. In the discussion section, it is mentioned this study uses the single-dose pharmacokinetic profile of cefiderocol from a study in healthy volunteers. In the study (ref. 18), a 60-minute infusion was used; however, cefiderocol is approved as a 3-hour infusion, thus some commentary on how this could impact the findings may be beneficial especially given that prolonged infusion of beta-lactams is later mentioned.  
  3. In the discussion section, immunomodulatory effects of other antimicrobials are specifically called out. Either in this section or the discussion section, consider commenting on whether other beta-lactams (or cephalosporins) have similar secondary effects as found in this study (beyond the referenced impact on interferon). 
  4. In relation to the previous comment, if cefiderocol is indeed the only cephalosporin which dampens the immune response, could one postulate a mechanistic relationship between this finding and the increased mortality seen in clinical trials? 

Author Response

Reviewer #1

Point 1: The authors should consider adding 1-2 sentences explaining the rationale for investigating cefiderocol specifically. 

Response: we have now added the rationale for this investigating, mentioning the CREDIBLE-CR study, which reported an increased all-cause mortality in the cefiderocol treatment group (relating to your comment#4).

Point 2: In the discussion section, it is mentioned this study uses the single-dose pharmacokinetic profile of cefiderocol from a study in healthy volunteers. In the study (ref. 18), a 60-minute infusion was used; however, cefiderocol is approved as a 3-hour infusion, thus some commentary on how this could impact the findings may be beneficial especially given that prolonged infusion of beta-lactams is later mentioned. 

Response: we have added some commentary in the discussion.  

Point 3: In the discussion section, immunomodulatory effects of other antimicrobials are specifically called out. Either in this section or the discussion section, consider commenting on whether other beta-lactams (or cephalosporins) have similar secondary effects as found in this study (beyond the referenced impact on interferon). 

Response: we have now elaborated this and included a recent study on cefazolin (a first-generation cephalosporin), which has been described to modulate pro-inflammatory response modulated by IL-2. Indeed, data on the secondary effects of beta-lactams/cephalosporin are scarce. We have also included some citations on the ambivalent nature of beta-lactams for a non-biased perspective on this matter.

Point 4: In relation to the previous comment, if cefiderocol is indeed the only cephalosporin which dampens the immune response, could one postulate a mechanistic relationship between this finding and the increased mortality seen in clinical trials? 

Response: we think that this is a possible explanation. However, in vivo studies are still needed to confirm this in vitro finding. We have excluded this statement in the manuscript on purpose, since this is only a speculation at this time point.

Reviewer 2 Report

The results of cefiderocol obtained in vitro should be checked in experimental model of sepsis and the inflammatory responses (TNF, interleukins  etc.) have to be compared with the results found in vitro study.  

Author Response

Reviewer #2

The results of cefiderocol obtained in vitro should be checked in experimental model of sepsis and the inflammatory responses (TNF, interleukins  etc.) have to be compared with the results found in vitro study.  

Response: thank you for your comments. We agree, which is why we have included this as a limitation in the discussion section. We believe that our findings, despite the preliminary nature, may be of interest for others to encourage research on the immunomodulatory or secondary effect of antibiotic therapy.

Reviewer 3 Report

The authors of this article entitled ‘‘Cefiderocol protects against cytokine- and endotoxin-induced disruption of vascular endothelial cell integrity in an in vitro experimental model’’ aimed to provide insight into the biological actions of cefiderocol underlying the pathophysiological mechanisms of sepsis, by exploring the possible effects of cefiderocol on the integrity of vascular endothelial cells during inflammation in an in vitro experimental model. The main strength of this study is its novelty in terms of attempting to shed light on the impact of cefiderocol with regard to the biological processes of the host that take place during sepsis, other than its antimicrobial effect. One of its limitations, as highlighted by the authors, is the fact that data are based on an in vitro experimental model and cannot be generalized, since they may not be applicable to in vivo models. In this context, the results should be interpreted with caution, since they represent preliminary findings based on an in vitro experimental model and need to be reproduced and confirmed in further studies. Overall it is a well written and organized manuscript. Below, several remarks are presented to the authors.

  1. When using abbreviations and acronyms, they should first be presented in the expanded form and abbreviated thereafter. This should be done separately for the abstract and for the main text. e.g. Abstract: LPS, IL-6, IL-1β, PBMCs. Text: TNF-α, IL-6, IL-1β, LPS, PBMCs, ELISA, TEER.

  1. In a certain part of the manuscript, there is lack of coherence in terms of writing flow. e.g. In the abstract the sentence ‘‘Stimulation with TNF-α and LPS increased the endothelial permeability assessed by TEER at 24 and 48 hours of co-culture’’ should immediately follow the sentence relating to the endothelial integrity, namely ‘‘ Endothelial integrity was quantified using transendothelial electrical resistance (TEER) measurement, performed at 0, 3, 24 and 48 hours after stimulation.’’

  1. In the legend of Figure 1 what is the meaning of ‘‘further unstimulated PBMCs’’?

  1. In Figure 1 what does n>/=3 represent in the sentence ‘‘Shown is mean and standard deviation of n>/=3.’’?

  1. In Figure 1, the diagrams provided show only one asterisk (*) with regard to statistical significance, but the legend below provides explanation as to what 1 vs 2 asterisks stand for. Where are the 2 asterisks in the diagram? Please revise.

  1. Why were 3 days chosen as the time period for the stimulation of PBMCs and not a different time window?

  1. The same question for the time period selected applies to the sentence ‘’vEC permeability was induced using 100 ng/ml TNF-α or 100 ng/ml LPS, known stimuli to disrupt vEC integrity [7, 8], in the presence and absence of cefiderocol (c=70 mg/L) for 48 hours’’. Why were 48 hours chosen and not a different time window? Please provide any relevant references.

  1. In the results section, when presenting the results that were statistically significant, p values should be provided for each separate result in parenthesis in the main text as well, and not only in the diagrams.
  2. ‘‘LPS pronouncedly induced all three cytokines.’’ Please provide p values for each separate cytokine. For TNF-α, although it is depicted that there is an increase after stimulation with LPS in Figure 1C, no p value is provided to support that the increase of TNF-α was significant. In case the TNF-α increase did not reach statistical significance, the above mentioned sentence should be revised.
  3. ‘‘Based on this data and a half-life of around 2 hours for cefiderocol [17], we have chosen 70 mg/L as the highest concentration for our in vitro experimental set-up. The lowest concentration was chosen based on the 4 times the MIC as a target minimum concentration close to 16 mg/L (based on the EUCAST breakpoint v11.0 for cefiderocol of ≥ 2 mg/L).’’ The explanation for the concentrations selected is rather vague to me and not completely clear. Could you be more detailed?
  4. Certain parts of the described methods may need additional clarification in order to become more reader friendly and easily comprehensible to a wider range of readers, for example to clinicians who are not adequately acquainted with these methods. For instance, how does co-culture of human umbilical vein cells with erythrocyte depleted whole blood mimic the physiological conditions in the blood vessel, as mentioned in section 2.2? It would be nice to explain the need to use erythrocyte depleted whole blood instead of whole blood for the particular in vitro experimental model.
  5. The phrase ‘‘antimicrobial substance with stability against metallo-beta-lactamase producing Gram-negative bacilli’’ needs to be rephrased. The way it is currently written suggests that stability is against bacilli, whereas stability is against metallo-beta-lactamase produced by Gram-negative bacilli.

  1. Please correct the phrase ‘‘there are increasing evidence’’ to ‘‘there is increasing evidence’’.

  1. Please revise the verb ‘‘mediated’’ in the sentence ‘‘a significant cause of mortality in infection-associated sepsis is mediated by pathophysiological processes related to vascular permeability’’.

  1. The sentence ‘‘To investigate, whether cefiderocol has an effect on the release of vascular-leakage-associated pro-inflammatory cytokines, TNF-α, IL-6 and IL-1β from LPS-stimulated PBMCs +/- cefiderocol was quantified by ELISA’’ needs several amendments to become easily understood. Comma after the word investigate should be omitted, vascular-leakage-associated should be rephrased, +/- should be replaced with words, was quantified should be in plural since I reckon it refers to TNF-α, IL-6 and IL-1β.

  1. Suggestion to use the word ‘‘markedly’’ instead of ‘‘pronouncedly’’.

Author Response

Reviewer #3

The authors of this article entitled ‘‘Cefiderocol protects against cytokine- and endotoxin-induced disruption of vascular endothelial cell integrity in an in vitro experimental model’’ aimed to provide insight into the biological actions of cefiderocol underlying the pathophysiological mechanisms of sepsis, by exploring the possible effects of cefiderocol on the integrity of vascular endothelial cells during inflammation in an in vitro experimental model. The main strength of this study is its novelty in terms of attempting to shed light on the impact of cefiderocol with regard to the biological processes of the host that take place during sepsis, other than its antimicrobial effect. One of its limitations, as highlighted by the authors, is the fact that data are based on an in vitro experimental model and cannot be generalized, since they may not be applicable to in vivo models. In this context, the results should be interpreted with caution, since they represent preliminary findings based on an in vitro experimental model and need to be reproduced and confirmed in further studies. Overall it is a well written and organized manuscript. Below, several remarks are presented to the authors.

Response: thank you for the thorough review and the helpful comments to improve our manuscript. We hope to have addressed your comments adequately.

Point 1: When using abbreviations and acronyms, they should first be presented in the expanded form and abbreviated thereafter. This should be done separately for the abstract and for the main text. e.g. Abstract: LPS, IL-6, IL-1β, PBMCs. Text: TNF-α, IL-6, IL-1β, LPS, PBMCs, ELISA, TEER.

Response: we have added this to the abstract and the main text.

Point 2: In a certain part of the manuscript, there is lack of coherence in terms of writing flow. e.g. In the abstract the sentence ‘‘Stimulation with TNF-α and LPS increased the endothelial permeability assessed by TEER at 24 and 48 hours of co-culture’’ should immediately follow the sentence relating to the endothelial integrity, namely ‘‘ Endothelial integrity was quantified using transendothelial electrical resistance (TEER) measurement, performed at 0, 3, 24 and 48 hours after stimulation.’’

Response: thank you for your suggestions, we have now rephrased this section for better flow and clarity.

Point 3: In the legend of Figure 1 what is the meaning of ‘‘further unstimulated PBMCs’’?

Response: we apologize for this confusion, we have now rephrased the figure legend to clarify that the unstimulated (no TNF) negative control was incubated with the highest cefiderocol concentration of 70 mg/L.

Point 4: In Figure 1 what does n>/=3 represent in the sentence ‘‘Shown is mean and standard deviation of n>/=3.’’?

Response: we have now clarified this and rephrased the figure legend. The mean and standard deviation was calculated from at least three independent experiments (biological replicates).

Point 5: In Figure 1, the diagrams provided show only one asterisk (*) with regard to statistical significance, but the legend below provides explanation as to what 1 vs 2 asterisks stand for. Where are the 2 asterisks in the diagram? Please revise.

Response: we have now revised this and removed the 2 asterisks

Point 6: Why were 3 days chosen as the time period for the stimulation of PBMCs and not a different time window?

Response: After three days the LPS-stimulated induction of analyzed cytokines was most pronounced and the inhibitory effect of FDC most impressive. Therefore, we decided to choose this timepoint.

Point 7: The same question for the time period selected applies to the sentence ‘’vEC permeability was induced using 100 ng/ml TNF-α or 100 ng/ml LPS, known stimuli to disrupt vEC integrity [7, 8], in the presence and absence of cefiderocol (c=70 mg/L) for 48 hours’’. Why were 48 hours chosen and not a different time window? Please provide any relevant references.

Response: ~70% of sepsis mortality occurs within the first 3 days after onset. Vascular leakage is considered as one of the hallmarks of sepsis, which is linked with (multi-)organ dysfunction. Pro-inflammatory cytokine, especially the early pro-inflammatory cytokines, can induce vascular leakage. Therefore, we measured the cellular integrity at defined intervals for the first 48 hours. TEER was measured after 0, 3, 24, and 48 hours. This is a pretty broad window that analyzes short term as well as long term effects. A pronounced decrease in TEER was observed after 24h that further decreased until 48 hours of stimulation. As the FDC-mediated inhibition of stimuli-mediated loss of integrity was most pronounced after 48 hours, we further analyzed different FDC concentrations at this timepoint.

Point 8: In the results section, when presenting the results that were statistically significant, p values should be provided for each separate result in parenthesis in the main text as well, and not only in the diagrams.

Response: we have now added the p-values of the statistically significant results in the main text. P=0.0495 were rounded to p=0.05 and is explicitly mentioned in the methods/statistics section.

Point 9: ‘‘LPS pronouncedly induced all three cytokines.’’ Please provide p values for each separate cytokine. For TNF-α, although it is depicted that there is an increase after stimulation with LPS in Figure 1C, no p value is provided to support that the increase of TNF-α was significant. In case the TNF-α increase did not reach statistical significance, the above mentioned sentence should be revised.

Response: the p-value was added and the sentence was rephrased accordingly (see main text of R1)

Point 10: ‘‘Based on this data and a half-life of around 2 hours for cefiderocol [17], we have chosen 70 mg/L as the highest concentration for our in vitro experimental set-up. The lowest concentration was chosen based on the 4 times the MIC as a target minimum concentration close to 16 mg/L (based on the EUCAST breakpoint v11.0 for cefiderocol of ≥ 2 mg/L).’’ The explanation for the concentrations selected is rather vague to me and not completely clear. Could you be more detailed?

Response: we have added more detail in the discussion. “Based on this data, a half-life of around 2 hours for cefiderocol [17] and the recommended dose regimen for cefiderocol of 2000 mg every 8 hours per infusionem, we have chosen 70 mg/L (half of the cmax after single-dose (2000 mg) administration over 60 minutes per infusionem, ~78 mg/L) as the highest concentration for our in vitro experimental set-up with the assumption that at least 70 mg/L of cefiderocol plasma concentration is maintained for ~50% of the time between administrations.”

Point 11: Certain parts of the described methods may need additional clarification in order to become more reader friendly and easily comprehensible to a wider range of readers, for example to clinicians who are not adequately acquainted with these methods. For instance, how does co-culture of human umbilical vein cells with erythrocyte depleted whole blood mimic the physiological conditions in the blood vessel, as mentioned in section 2.2? It would be nice to explain the need to use erythrocyte depleted whole blood instead of whole blood for the particular in vitro experimental model.

Response: we have now rephrased and extended the description of the method to include a rationale for the depletion of red blood cells.

Point 12: The phrase ‘‘antimicrobial substance with stability against metallo-beta-lactamase producing Gram-negative bacilli’’ needs to be rephrased. The way it is currently written suggests that stability is against bacilli, whereas stability is against metallo-beta-lactamase produced by Gram-negative bacilli.

Response: done

Point 13: Please correct the phrase ‘‘there are increasing evidence’’ to ‘‘there is increasing evidence’’.

Response: done

Point 14: Please revise the verb ‘‘mediated’’ in the sentence ‘‘a significant cause of mortality in infection-associated sepsis is mediated by pathophysiological processes related to vascular permeability’’.

Response: done

Point 15: The sentence ‘‘To investigate, whether cefiderocol has an effect on the release of vascular-leakage-associated pro-inflammatory cytokines, TNF-α, IL-6 and IL-1β from LPS-stimulated PBMCs +/- cefiderocol was quantified by ELISA’’ needs several amendments to become easily understood. Comma after the word investigate should be omitted, vascular-leakage-associated should be rephrased, +/- should be replaced with words, was quantified should be in plural since I reckon it refers to TNF-α, IL-6 and IL-1β.

Response: thank you for this comment. We have now rephrased the whole sentence for better clarity.

Point 16: Suggestion to use the word ‘‘markedly’’ instead of ‘‘pronouncedly’’.

Response: done